# New Series of Red-Light Phosphor Ca_9−_*_x_*Zn*_x_*Gd_0.9_(PO_4_)_7_:0.1Eu^3+^ (*x* = 0–1)

**DOI:** 10.3390/molecules28010352

**Published:** 2023-01-01

**Authors:** Yury Yu. Dikhtyar, Dmitry A. Spassky, Vladimir A. Morozov, Sergey N. Polyakov, Valerya D. Romanova, Sergey Yu. Stefanovich, Dina V. Deyneko, Oksana V. Baryshnikova, Ivan V. Nikiforov, Bogan I. Lazoryak

**Affiliations:** 1Chemistry Department, Lomonosov Moscow State University, 119991 Moscow, Russia; 2Skobeltsyn Institute of Nuclear Physics, Lomonosov Moscow State University, 119991 Moscow, Russia; 3Institute of Physics, University of Tartu, W. Ostwald str. 1, 50411 Tartu, Estonia; 4Technological Institute for Superhard and Novel Carbon Materials, Troitsk, 108840 Moscow, Russia; 5Laboratory of Arctic Mineralogy and Material Sciences, Kola Science Centre, Russian Academy of Sciences, 14 Fersman str.,184209 Apatity, Russia

**Keywords:** phosphate, *β*-Ca_3_(PO_4_)_2_, *β*-TCP, red luminescence, crystal structure

## Abstract

In this study, a new series of phosphors, Ca_9−*x*_Zn*_x_*Gd_0.9_(PO_4_)_7_:0.1Eu^3+^ (*x* = 0.00–1.00, step d*x* 0.05), was synthesized, consisting of centro- and non-centrosymmetric phases with *β-*Ca_3_(PO_4_)_2_-type structure. Crystal structures with space groups *R*3*c* (0.00 ≤ *x* < 0.35) and *R*$\bar{3}$*c* (*x* > 0.8) were determined using X-ray powder diffraction and the method of optical second harmonic generation. In the region 0.35 ≤ *x* ≤ 0.75, phases *R*3*c* and *R*$\bar{3}$*c* were present simultaneously. Refinement of the Ca_8_ZnGd(PO_4_)_7_ crystal structure with the Rietveld method showed that 71% of Gd^3+^ ions are in *M*3 sites and 29% are in *M*1 sites. A luminescent spectroscopy study of Ca_9−*x*_Zn*_x_*Gd_0.9_(PO_4_)_7_:0.1Eu^3+^ indicated the energy transfer from the crystalline host to the Gd^3+^ and Eu^3+^ luminescent centers. The maximum Eu^3+^ luminescence intensity corresponds to the composition with *x* = 1.

## 1. Introduction

Structural type *β*-Ca_3_(PO_4_)_2_ (space group (SG): *R*3*c*, *Z* = 21) [1] is a promising base for the development of materials with various interesting properties: luminescent [2], laser [3], non-linear optical [4,5,6], ferroelectric [4,5,6], antiferroelectric [7,8], catalytic [9], bioregenerative [10] and antibacterial [11]. In particular, these compounds can be expected to be of practical interest as narrow-band red light phosphors for the creation of white-light emission diodes (WLEDs), or as bone substitute material with ability to use imaging techniques.

The creation of new synthetic bone substitutes is an important task due to the rapid growth of the aging population and the increasing number of people with diseases of the musculoskeletal system [12]. Ceramics, based on calcium phosphates, are the most common material for filling bone defects [13]. Of all possible calcium orthophosphates (hydroxyapatite Ca_10_(PO_4_)_6_(OH)_2_ [14], *β*-tricalcium phosphate *β*-Ca_3_(PO_4_)_2_ [15], *a*-tricalcium phosphate *a*-Ca_3_(PO_4_)_2_ [16], brushite CaHPO_4_·2H_2_O [17], octacalcium phosphate Ca_8_(HPO_4_)_2_(PO_4_)_4_·5H_2_O [18]), *β*-Ca_3_(PO_4_)_2_ is one of the most applicable for bone tissue restoration [19].

The development of a highly efficient red phosphor is an urgent task because commercially available WLED based on (Ga,In)N (465 nm) blue-chip [20] and bright-yellow phosphor Y_3_Al_2_(AlO_4_)_3_:Ce^3+^ (YAG:Ce^3+^) [21,22] suffers from several disadvantages: low color rendering index (CRI) [23,24,25,26], high correlated color temperature [27], bad influence on a person’s psychological state and health of the eyes [28,29]. Using the UV-chip [30] and varying blue, red [31,32,33] and green phosphors, a WLED with the necessary characteristics can be obtained [34,35].

The unique structure of *β*-Ca_3_(PO_4_)_2_ has wide possibilities for cationic substitution. For example, incorporation of Eu^3+^ in *β*-Ca_3_(PO_4_)_2_ makes it possible to obtain red-light phosphors, while co-doping of Eu^3+^ and Gd^3+^ allows for the acquirement of even more superior luminescent properties. In [36] it was shown that the Ca_9_Gd_0.1_Eu_0.9_(PO_4_)_7_ phosphor emitted red light by 4.13 times brighter than the commercially available Y_2_O_3_:Eu^3+^. In [37,38], the authors found that sample Ca_8_MgGd(PO_4_)_7_:Eu^3+^ had the highest luminescent intensity in comparison with Ca_8_MgY(PO_4_)_7_:Eu^3+^ and Ca_8_MgLa(PO_4_)_7_:Eu^3+^. This may be related to the sensibilization effect of Gd^3+^ in the energy transfer processes to Eu^3+^. Thus, the simultaneous incorporation of Eu^3+^ and Gd^3+^ can improve the luminescent characteristics of material (due to energy transfer processes from the matrix to Gd^3+^ and further to Eu^3+^-centers), and also can act as a contrast agent for MRI (magnetic resonance imaging) and X-ray dual imaging, which consists of combining two radiographs acquired at two different lanthanides [39].

Further improvement of luminescent properties is possible due to isovalent cation substitutions within the *β*-Ca_3_(PO_4_)_2_ host. This can also lead to a change in SG (supporting information in [40]), which should be taken into account. Sometimes, the authors did not define the crystal structure of Ca_8_*M*Gd(PO_4_)_7_:Eu^3+^ (*M*^2+^ = Zn, Mg, Cd) with SG *R*3*c* correctly. Substitution Ca^2+^ → Zn^2+^ in the *M*5 site of *β*-Ca_3_(PO_4_)_2_ leads to the change in SG *R*3*c* → *R*$\bar{3}$*c*, the improvement of materials’ luminescent properties (the explanation of this phenomena is provided in detail in [2,41]) and antibacterial characteristics [42] (Figure 1). Given the highest luminescent characteristics of phosphor with *M* = Zn^2+^ in Ca_8_*M*Eu(PO_4_)_7_ [41] (*M* = Ca^2+^, Mg^2+^, Zn^2+^, Cd^2+^), it can be assumed that Ca_8_ZnGd(PO_4_)_7_:Eu^3+^ will show improved luminescent properties.

In the present study, a number of solid solutions with the general formula Ca_9−*x*_Zn*_x_*Gd_0.9_Eu_0.1_(PO_4_)_7_ (*x =* 0.00–1.00, d*x* = 0.05) are synthesized and examined for the first time. The following questions were in the focus of the study: (1) the boundary of the two-phase region with the change of SG *R*3*c* → *R*$\bar{3}$*c* at gradual substitution Ca^2+^ → Zn^2+^; (2) the distribution of cations in the Ca_8_ZnGd(PO_4_)_7_ structure; (3) the modification of luminescence properties, in particular the energy transfer to Eu^3+^ with crystal compositions.

## 2. Results and Discussions

### 2.1. Elemental Composition and Preliminary Characterization

The quantitative ratio of elements was determined by EDX analysis. The results for samples with x = 0.35, 0.50, 0.75 and 1.00 in Ca_9−*x*_Zn*_x_*Gd_0.9_Eu_0.1_(PO_4_)_7_ series show the insignificant deviation from the theoretical composition. Table 1 summarizes the results of the EDX analysis.

### 2.2. Scanning Electron Microscopy

Figure 2 shows the images obtained with the SEM method for x = 0.00, 0.35, 0.75 and 1.00 in the Ca_9−*x*_Zn*_x_*Gd_0.9_Eu_0.1_(PO_4_)_7_ series. The shape of the particles becomes sharper, and the particles form larger agglomerates under Ca*^2+^*→ Zn*^2+^* substitution in Ca_9__−*x*_Zn*_x_*Gd_0.9_Eu_0.1_(PO_4_)_7_. This is correlated with the transition from the non-centrosymmetric to centrosymmetric (R3c → R$\bar{3}$c) state.

### 2.3. PXRD Analysis

A number and peaks position of synthesized solid solutions in Ca_9−*x*_Zn*_x_*Gd(PO_4_)_7_:0.1Eu^3+^ PXRD patterns related to similar compounds with *β*-Ca_3_(PO_4_)_2_-type structure [2]. The absence of impurity phases shows that Gd^3+^ (*r*_VI_ = 0.94 Å), Eu^3+^ (*r*_VI_ = 0.95 Å) and Zn^2+^ (*r*_VI_ = 0.74 Å [43]) ions are completely incorporated into the *β*-Ca_3_(PO_4_)_2_-type structure.

The unit cells parameters decrease with increasing of Zn^2+^ concentration because the ionic radius of Zn^2+^ (*r*_VI_ = 0.74 Å) is smaller than that of Ca^2+^ (*r*_VI_ = 1.00 Å) (Appendix A). However, this decrease in parameters is nonlinear. Figure 3a shows that the slope of the curve in the region of 0.00 ≤ *x* < 0.30 is less than in the region of 0.30 ≤ *x* ≤ 0.80, and there is a sharp jump in the change of parameters for the composition *x* = 0.50. Such comprehensive behavior of unit cell parameters on Zn^2+^ concentration can be explained by a change in the SG *R*3*c* → *R*$\bar{3}$*c*. In a routine laboratory experiment, PXRD patterns of compounds with these SGs are indistinguishable [44,45].

In the region *x* = 0.35–0.75, a decrease in the crystalline size and an increase in the FWHM in comparison with *x* = 1 is observed. This circumstance may also indicate the coexistence of two phases—*R*3*c* and *R*$\bar{3}$*c*—further confirmed by the SHG method.

### 2.4. SHG Study

The above-defined limitations on the existence of regions with different symmetries are consistent with the SHG data (Appendix A). As Zn^2+^ ions concentration rises from 0 to 0.35, there is a slight decrease in SHG signal (Figure 3b). More rapid and nonmonotonic decrease of SHG is observed in an interphase region with 0.35 < *x* < 0.75, where centro- and non-centrosymmetry fragments of whitlockite-like structures are mixed. Very small SHG activity at 0.75 < *x* ≤ 1.00 corresponds to centrosymmetric phase (SG *R*$\bar{3}$*c*) distorted with defects near the smaller boundary of *x*. For *x* > 0.75, the SHG signal is null or at the background level in accordance with macroscopic center of symmetry in this phase.

### 2.5. Crystal Structure Refinement of Ca_8_ZnGd(PO_4_)_7_

Compounds with a *β*-Ca_3_(PO_4_)_2_-type structure have extended possibilities for cationic substitution. There are six crystallographic positions with different sizes *M*1–*M*5 and *M*6 for Ca^2+^ in *β*-Ca_3_(PO_4_)_2_. The occupancy of *M*4 site can vary from 0 to 1, while *M*6 is always vacant. Large numbers of positions and vacancies suggests a wide opportunity for iso- and heterovalent cationic substitution, including the lanthanoids *Ln*^3+^. These substitutions may lead to changing of SG from polar to non-polar (Supplementary information of [40]).

Atomic coordinates of phosphate Ca_8_ZnLa(PO_4_)_7_ (SG *R*$\bar{3}$*c*) [46] were used as an initial model for the structural refinement of Ca_8_ZnGd(PO_4_)_7_. The *M*1 and *M*3 sites (36*f*) are jointly occupied by La^3+^ and Ca^2+^; *M*5 (6*b*) is occupied by Zn^2+^ in Ca_8_ZnLa(PO_4_)_7_. There is no *M*2 site in this structure with *R*$\bar{3}$*c* SG, since *M*2 is equivalent to *M*1 (*M*1 = *M*2). In centrosymmetric model with *R*$\bar{3}$*c* SG, the name of the *M*3 site was left the same, as for the polar model with *R*3*c* SG of *β*-TCP-type structure. There are two phosphorus atoms in 12*c* and 36*f* Wyckoff positions. There is one oxygen atom in 12*c* Wyckoff position, and the other five are in 36*f* positions.

At the first step of the refinement, the *f*-curves for Ca^2+^ (in *M*1 and *M*3 sites) and Zn^2+^ (in *M*5) were used to form the determination of the atoms’ positions. This analysis (Table 2) shows that the Gd^3+^ ions are distributed between the positions *M*1, *M*3 (exceeding the maximum occupancy—1 for *M*1 and 0.5 for *M*3), while the *M*5 position is completely occupied by Zn^2+^ ions.

The *M*3 and P1 sites must be in special Wyckoff positions (18*d*) (0.5, 0, 0) and (6*a*) (0, 0, 0.25), respectively, in Ca_8_ZnGd(PO_4_)_7_ (SG *R*$\bar{3}$*c*). The structure refinement of this model led to large parameters of atomic displacement, *U*_iso_. = 0.162(2) Å^2^ for Ca^2+^/Gd^3+^ in the *M*3 site, and *U*_iso_. = 0.173(4) Å^2^ for P1. For this reason, the refinement of the Ca_8_ZnGd(PO_4_)_7_ structure was performed with a shift of the phosphorus atom P1 from a special (6*a*) position to a half-occupied (12*c*) position. The Ca^2+^ in *M*3 was shifted from a special (18*d*) position to a half-occupied (36*f*) site. Moreover, the positions of Ca^2+^ and Gd^3+^ in *M*1 and *M*3 sites were additionally split. This led to a significant decrease in *M*1 and *M*3 *U*_iso_.

After refinement, there is a good agreement between the calculated and experimental X-ray diffraction patterns (Figure 4) with acceptable *R*-factors (Table 2). Fractional atomic coordinates, site symmetry, isotropic displacement of atomic parameters and site occupation for Ca_8_ZnGd(PO_4_)_7_ are shown in Appendix A. The main interatomic distances are shown in Appendix A. The distribution of Gd^3+^ ions in Ca_8_ZnGd(PO_4_)_7_ over crystal sites was found to be 71% in *M*3, and 29% in *M*1. In Ca_8_MgGd(PO_4_)_7_ [47], the distribution of Gd^3+^ ions was 77% in *M*3, and 23% in *M*1 sites. Thus, the *M*3 site is a preferable location for relatively big Gd^3+^ (*r*_VI_ = 0.94 Å) ions.

The average length of the *M*-O bonds in the *M*1O_9_ and *M*3O_8_ polyhedra are *d*_<*M*1-O>_ = 2.347 Å and *d*_<*M*3-O>_ = 2.526 Å, respectively. At the same time, the deviations of *M*-O distances from the average values of *d*_<*M*1-O>_ and *d*_<*M*3-O>_ are significant and indicate a large distortion of the polyhedra. Calculated distortion indexes (*DI* [48]) for *M*1O_9_ and *M*3O_8_ polyhedra are *DI_M_*_1O_ = 0.0362 and *DI_M_*_3O_ = 0.0563. These values were calculated according to the Equation (1)
(1)$$DI=\frac{1}{n}\overset{n}{\underset{i=1}{\sum }}\frac{d_{i}}{\langle d\rangle }$$
where *n*—number of bonds, *d_i_*—length of a bond and <*d*>—average bond length for polyhedra. In Ca_9_Gd(PO_4_)_7_, *DI*’s for *M*1O_8_, *M*2O_8_ and *M*3O_8_ are *DI_M_*_1O_ = 0.0486, *DI_M_*_2O_ = 0.0234 and *DI_M_*_3O_ = 0.0500 (*M*1 and *M*2 sites in SG *R*$\bar{3}$*c* are equivalent) [49]. It indicates that *M*3 site (77% Gd^3+^) in Ca_8_ZnGd(PO_4_)_7_ (SG *R*$\bar{3}$*c*) is more distorted than *M*3 site in Ca_9_Gd(PO_4_)_7_ (SG *R*3*c*).

### 2.6. Luminescent Properties of Ca_9−x_Zn_x_Gd(PO_4_)_7_:0.1Eu^3+^

The luminescence spectra of Ca_9−*x*_Zn*_x_*Gd_0.9_Eu_0.1_(PO_4_)_7_ (*x* = 0, 1) compounds are presented in Figure 5a,b at 6 and 300 K. The spectra consist of a set of narrow emission lines in the region of 305–317 nm and 575–720 nm, related to 4*f*-4*f* transitions in Gd^3+^ and Eu^3+^, respectively. The features of the structure of Eu^3+^ emission change with the incorporation of Zn^2+^. It can be shown for the ^5^D_0–_^7^F_0_ transition (Figure 6), which is not split by the crystal field and sensitive to the number of nonequivalent Eu^3+^ positions. Two peaks can be observed for the sample with *x* = 0, while only one for *x* = 1. The number of peaks correlate with the decrease of the number of cationic positions. However, presence of slightly different crystallographic sites with similar crystallographic properties may also result in a single peak consisting of several superimposed peaks. For *x* = 0 (SG *R*3*c*), there are three cationic positions—*M*1, *M*2 and *M*3, while for *x* = 1 (SG *R*$\bar{3}$*c*) there are only two cationic positions—*M*1 and *M*3. Analysis of the emission spectra in the 305–317 nm region demonstrates that the number of peaks related to Gd^3+^ also depends on the crystal composition. Presence of additional features such as the shoulder at 310.6 nm and peak at 314 nm depends on the Zn^2+^ content in the sample. This is especially demonstrative when the temperature drops to 6 K (Figure 5b)—the peak at 314 nm becomes more prominent for the compound with *x* = 0.

The relative intensities of Gd^3+^ and Eu^3+^ emissions depend on the Zn^2+^ content in the sample under VUV excitation (163 nm). The given wavelength is tentatively related to the fundamental absorption region [2], and observed modifications are connected with the features of energy transfer to competitive emission centers. It was found that in the sample with *x* = 0, the Gd^3+^ luminescence band dominated in the spectrum, while for *x* = 1, the Eu^3+^ luminescence band intensity increased (Figure 5). Therefore, the energy transfer from the host to Eu^3+^ is more efficient in the sample with Zn^2+^.

PLE spectra of Eu^3+^ and Gd^3+^ emissions are presented in Figure 7. In the excitation spectra of Eu^3+^ a set of narrow lines in the region of 320–500 nm is connected with 4*f*-4*f* Eu^3+^ transitions, while the broad band peaking at ~245 nm to charge transfer transitions from the valence band (VB) to 4*f* Eu states. A narrow excitation band at 273 nm is related to ^8^S_7/2_–^6^I*_J_* transitions in Gd^3+^, thus indicating the energy transfer from Gd^3+^ to Eu^3+^. The scheme of Eu^3+^ and Gd^3+^ energy levels position relative to the top of the valence and bottom of the conduction bands is presented in Figure 8. The position of Eu^3+^ and Gd^3+^ 4*f* ground states position were taken from [2], while 4*f* excited states were taken from a Dieke diagram [50]. In the excitation spectra of Gd^3+^ emissions, a set of narrow lines is observed at 246, 253 and 273 nm, which are connected with ^8^S_7/2_–^6^D*_J_* and ^8^S_7/2_–^6^I*_J_* Gd^3+^ transitions. A very weak broad band can be found in the region of 210–250 nm. The position of this band coincides with the position of the charge-transfer band (CTB) in the excitation spectra of Eu^3+^ and shows that the energy transfer from Eu^3+^ to Gd^3+^ is possible as well; however, its efficiency is low. Similarity of the PLE of Ca_9_Gd_0.9_Eu_0.1_(PO_4_)_7_ and Ca_8_ZnGd_0.9_Eu_0.1_(PO_4_)_7_ in the energy range up to the fundamental absorption region indicates that electronic states of Zn^2+^ do not form additional channels of energy transfer to the emission centers. In the region of the fundamental absorption edge, the excitation spectra of the studied samples differ considerably. In the sample with Zn^2+^, a broad band peaking at 157 nm (7.89 eV) is observed. The obtained value corresponds to the energy of the direct creation of excitons in Ca_8_Zn*Ln*(PO_4_)_7_ compounds—7.94 eV [2]. Therefore, we also attribute the peak at 7.89 eV to the direct exciton creation in Ca_9−*x*_Zn*_x_*Gd_0.9_Eu_0.1_(PO_4_)_7_ (*x* = 0, 1). A small hump can also be found at 172 nm (7.21 eV) in the excitation spectrum of Eu^3+^ for the sample with *x* = 1, while this peak dominates in the excitation spectra of both Eu^3+^ and Gd^3+^ emissions in the sample with *x* = 0. Previously, a set of sharp peaks related to 4f-5d transitions in Eu^3+^ were observed in the VUV spectral region in some wide bandgap compounds [51]. A similar origin could be supposed for the detected peak at 172 nm. However, this sharp peak is intensive in the excitation spectra of Gd^3+^ emission, thus indicating efficient energy transfer from Eu^3+^ to Gd^3+^. However, according to the analysis of the excitation spectra in UV spectral region such kind of energy transfer is inefficient (CTB is barely observed in the excitation spectra of Gd^3+^ emission) and therefore the attribution of the peak at 172 nm to 4*f*-5*d* transitions in Eu^3+^ is low probable. This peak can be tentatively ascribed to excitons localized near Gd^3+^. This may be the reason for the enhanced energy transfer from the host to Gd^3+^ in this compound. In the sample with Zn^2+^, localization of excitons near Gd^3+^ is less effective, which results in the increase in intensity of Eu^3+^ luminescence increases.

## 3. Experimental Section

### 3.1. Sample Preparation

Ca_9−*x*_Zn*_x_*Gd_0.9_Eu_0.1_(PO_4_)_7_ (*x =* 0.00–1.00, d*x* = 0.05) compounds were synthesized by solid-state method from stoichiometric mixtures of CaHPO_4_·2H_2_O (99.9%), CaCO_3_ (99.9%), ZnO (99.9%) and Ln_2_O_3_ (*Ln*^3+^ = Gd, Eu) (99.9%), purchased from Sigma-Aldrich, according to the reaction:$$7CaHPO_{4}\cdot 2H_{2}O+\left(2-x\right)CaCO_{3}+xZnO+0.45Gd_{2}O_{3}+0.05Eu_{2}O_{3}\;\;\;\;\;\;\;\;=Ca_{9-x}Zn_{x}Gd_{0.9}Eu_{0.1}{\left(PO_{4}\right)}_{7}+17.5H_{2}O+\left(2-x\right)CO_{2}$$

All reagents were controlled by PXRD for the absence of impurity phases. The stoichiometric mixtures were carefully grounded and very slowly heated up to 600 K for 9 h and then annealed at 1273 K for 10 h with several intermediate grindings followed by slow cooling (10 h) to room temperature (T_R_).

### 3.2. Experimental Description

The powder X-ray diffraction (PXRD) patterns were collected on a Thermo ARL X’TRA powder diffractometer (CuK_α_ radiation, λ = 1.5418 Å, Bragg–Brentano geometry, scintillator detector) at T_R_ in 2θ range of 5–65° with steps of 0.02°. The phase analysis of the obtained samples was carried out using the Crystallographica Search-March program (Version 2.0.3.1) and JCPDS PDF-2 database. Le Bail analysis was performed using JANA2006 program package [52]. The Debye–Scherrer equation [53] was implemented to count coherent scattering regions (crystalline sizes). LaB_6_ (SRM 660c) as a line shape standard was applied to determine instrumental broadening.

Scanning electron microscopy (SEM) study was performed using Tescan VEGA3 microscope equipped with LaB_6_ cathode. The SEM images were obtained using secondary electron detector. The analysis of the quantitative of elements concentration was determined by energy-dispersive X-ray (EDX) analysis. Ca_K_, Eu_L_, Gd_L_ and Zn_L_ lines in the EDX spectra were used for the element content determination.

An Empyrean X-ray diffractometer (PANalytical, Almelo, Netherlands) equipped with a PIXcel^3D^ 2D solid-state hybrid detector providing for counting photons with a high spatial resolution and a high dynamic range was used for registration a powder diffraction patterns. Each pixel is 55 μm × 55 μm and detector array is 256 × 256 pixels. Bragg–Brentano geometry was realized with a Bragg—Brentano^HD^ X-ray optical module (parabolic multilayer mirror), which monochromatized the primary X-ray beam and provided it with high intensities compare to the commonly used divergence slits and beta filters as well as increase the peak-to-background ratio and minimize excitation of fluorescent radiation from the sample. The X-ray generator (CuK*_α_*-radiation) was operated at 40 kV and 40 mA. Diffraction patterns were recorded in the range of 10÷110^0^ (2θ) with the step size of 0.0131^0^ using continuous scan mode. PIXcel^3D^ operated in a scanning line detector (1D) mode over its total active length (14 mm), which corresponds 3.3^0^ (2θ) on the Empyrean goniometer with the radius of 240 mm. To avoid an influence of transparency effect of a material, we used a zero-background sample holder consisting of an obliquely cut silicon single crystal with a 32 mm diameter and 2 mm thickness. Rietveld analysis [54] was performed using Jana2006 program package. Illustrations were made using the VESTA program.

The second harmonic generation signal was measured with a Q-switched YAG:Nd laser at λ_ω_ = 1064 nm in reflection mode for powder series with particle sizes of 40–60 μm. The laser operated with a repetition rate of 10 impulses/s and an impulse duration of about 5 ns. The laser beam was split into two beams to excite the radiation at the halved wavelength, λ_2ω_ = 532 nm, simultaneously in samples to be measured and in reference sample polycrystalline *α*-SiO_2_. The incident beam peak power was about 0.1 mW on a spot 3 mm in diameter on the surface of the sample.

Photoluminescence emission (PL) and excitation (PLE) spectra were measured using specialized setups for ultraviolet (UV) and vacuum ultraviolet (VUV) luminescence spectroscopy. A DDS-400 deuterium lamp was used as an excitation source for measurements in UV-visible spectral regions (200–500 nm). Excitation wavelengths were monochromatized using primary prism monochromator DMR-4. PLE spectra were measured with 5 nm spectral resolution. The sample was installed in a Janis VPF-800 nitrogen cryostat, which allows for changing the temperature in the range of 80–700 K. The luminescence signal was registered using an ARC SpectraPro-300i monochromator and a Hamamatsu H28259-01 photon counting head used for the measurements of luminescence.

The measurements in UV-VUV spectral region (130–400 nm) were performed using deuterium lamp Hamamatsu L11798 with MgF_2_ output windows as an excitation source. The lamp radiation was monochromatized using a McPherson 234/302 vacuum primary monochromator. The PLE spectra were measured with 14 nm resolution. The samples were placed in an ARS vacuum cryostat, which allows for measurements in the temperature range of 5–300 K. Luminescence was registered using Shamrock 303i monochromator equipped with ANDOR iDUS CCD detector with a spectral resolution of 0.3 nm.

## 4. Conclusions

The as-synthesized Ca_9-*x*_Zn*_x_*Gd_0.9_Eu_0.1_(PO_4_)_7_ solid solutions with *β*-Ca_3_(PO_4_)_2_-type structures do not contain impurity phases and all crystallize in *R*3*c* or *R*$\bar{3}$*c* SGs depending on the content of Zn^2+^ ions. Centrosymmetry in Ca_9−*x*_Zn*_x_*Gd_0.9_Eu_0.1_(PO_4_)_7_ phosphors with *x* > 0.75 was reliably established based on the absence of the SHG effect. Together with data from X-ray analysis, this proves the existence of a pure centrosymmetric variant of the *β*-Ca_3_(PO_4_)_2_-type compounds with *R*$\bar{3}$*c* SG and the interphase region between this phase and compounds with polar SG *R*3*c* at lower *x*. It is worth noting that SGs *R*3*c* and *R*$\bar{3}$*c* are practically indistinguishable in routine laboratory X-ray diffraction experiments because the regions of centro- and non-centrosymmetric phases can only be definitely determined using additional methods such as SHG. Namely, in a Ca_9−*x*_Zn*_x_*Gd_0.9_Eu_0.1_(PO_4_)_7_ system, the change of symmetry was confirmed by this method. Concentration phase boundaries in this system were established according to zero (or very small) SHG signal for the *R*$\bar{3}$*c* phase in interval 0.80 < *x* ≤ 1.00. Then, at 0.35 ≤ *x* ≤ 0.75 a transient two-phase state was observed, and at 0.00 ≤ *x* < 0.35 a non-centrosymmetric SG *R*3*c* was fully stabilized according to nearly constant non-zero SHG signal.

In the single-phase phosphate Ca_8_ZnGd(PO_4_)_7_, it was shown using the Rietveld method that Zn^2+^ ions occupy the *M*5 position, thus relieving a geometric stress in the structure. The *M*3 position is shifted from the special position (18*d*) to the semi-occupied position (36*f*), and P1 position is shifted from the special position (6*a*) to the semi-occupied general position (12*c*). The rationale for these shifts was the large values of the U_iso_ parameters (atomic displacement) in refining the structure under the assumption of locating *M*3 and P1 in special positions. Thus, the *M*3 position is shifted from the third-order axis, and its occupancy is equal to 0.5. The distribution of Gd^3+^ ions in Ca_8_ZnGd(PO_4_)_7_ by position in the structure turned out to be 71% in *M*3 and 29% in *M*1. This is characteristic of large cations such as Gd^3+^. Thus, we can distinguish the following factors, which probably positively affect the intensity of Eu^3+^ luminescence:
(1)Shifting of the *M*3 position from the third-order axis;(2)Distortion of *M*3O_8_ polyhedra (local decrease of the symmetry);(3)General increase in symmetry of the structure (*R*3*c* → *R*$\bar{3}$*c*).

The relative intensity of Eu^3+^/Gd^3+^ emissions depend on their composition. It is shown that in compounds with *x* = 1 the energy transfer from the host to Eu^3+^ is improved, which results in the increase in Eu^3+^ luminescent intensity.

Two nonequivalent positions of *Ln*^3+^ ions were deduced from the emission spectra of Eu^3+^ as well as Gd^3+^ in the sample with *x* = 0. Therefore, Gd^3+^ ions could also be used as a luminescent marker to study the crystal structure of the compound.

## Figures and Tables

**Figure 1 molecules-28-00352-f001:**
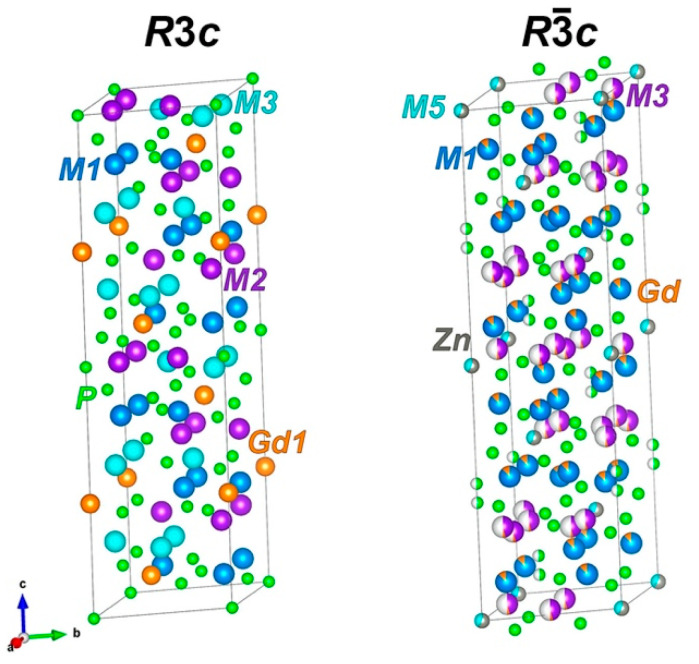
*β*-Ca_3_(PO_4_)_2_-type structure with *R*3*c* (Ca_9_Gd(PO_4_)_7_) and *R*$\bar{3}$*c* (Ca_8_ZnGd(PO_4_)_7_) space groups. Half of the spheres show occupancy of 0.5 (for example, in *M*3 site).

**Figure 2 molecules-28-00352-f002:**
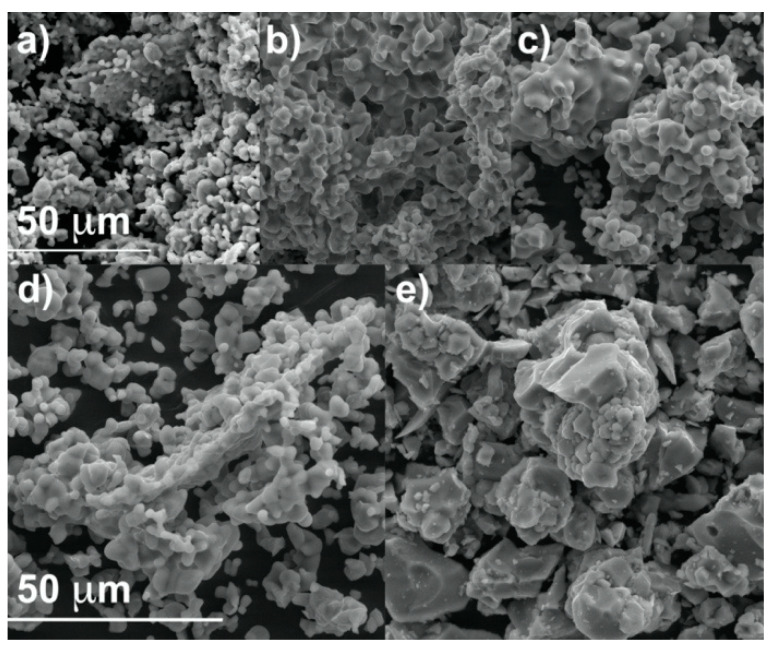
The SEM images for x = 0.00 (**a**), 0.35 (**b**), 0.50 (**c**), 0.75 (**d**) and 1.00 (**e**) in Ca_9−*x*_Zn*_x_*Gd_0.9_Eu_0.1_(PO_4_)_7_ series.

**Figure 3 molecules-28-00352-f003:**
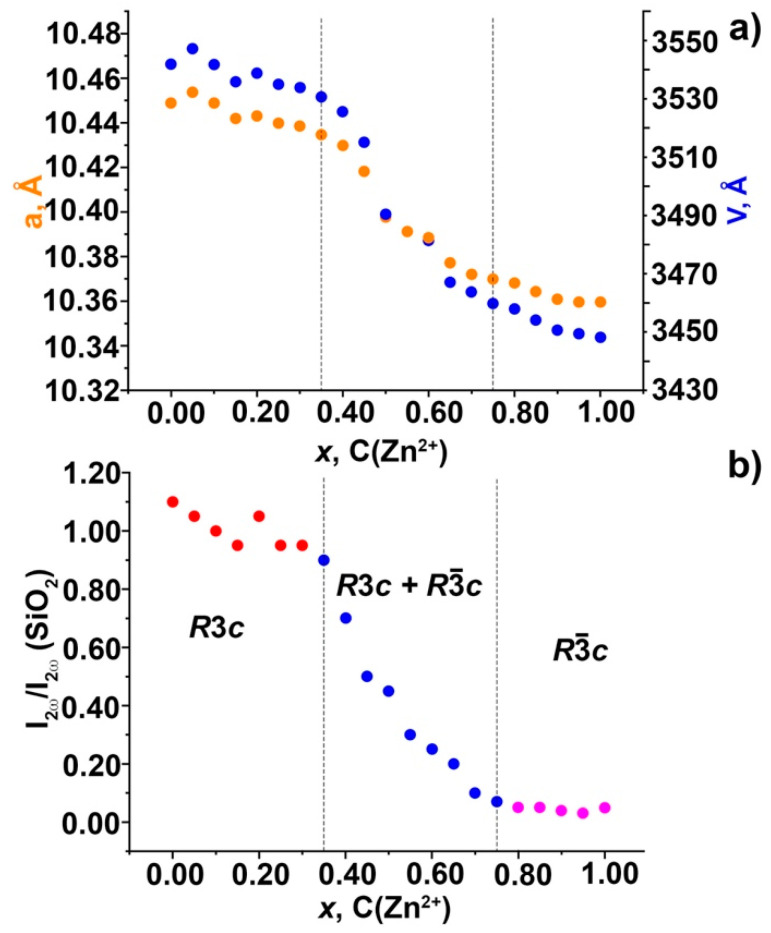
Dependence of unit cell parameters (**a**) and the second harmonic generation signal (*I*_2ω_/*I*_2ω_(SiO_2_)) (**b**) on the Zn^2+^ concentration for Ca_9−*x*_Zn*_x_*Gd(PO_4_)_7_:0.1Eu^3+^ solid solutions.

**Figure 4 molecules-28-00352-f004:**
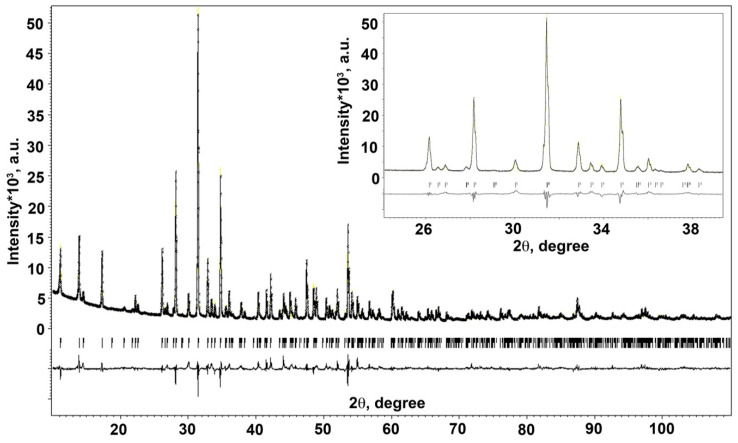
Fragment of observed, calculated and difference PXRD data for Ca_8_ZnGd(PO_4_)_7_. Tick marks denote the peak positions of possible Bragg reflections. The inset shows detailed fragment of PXRD data for 2θ = 24–40°.

**Figure 5 molecules-28-00352-f005:**
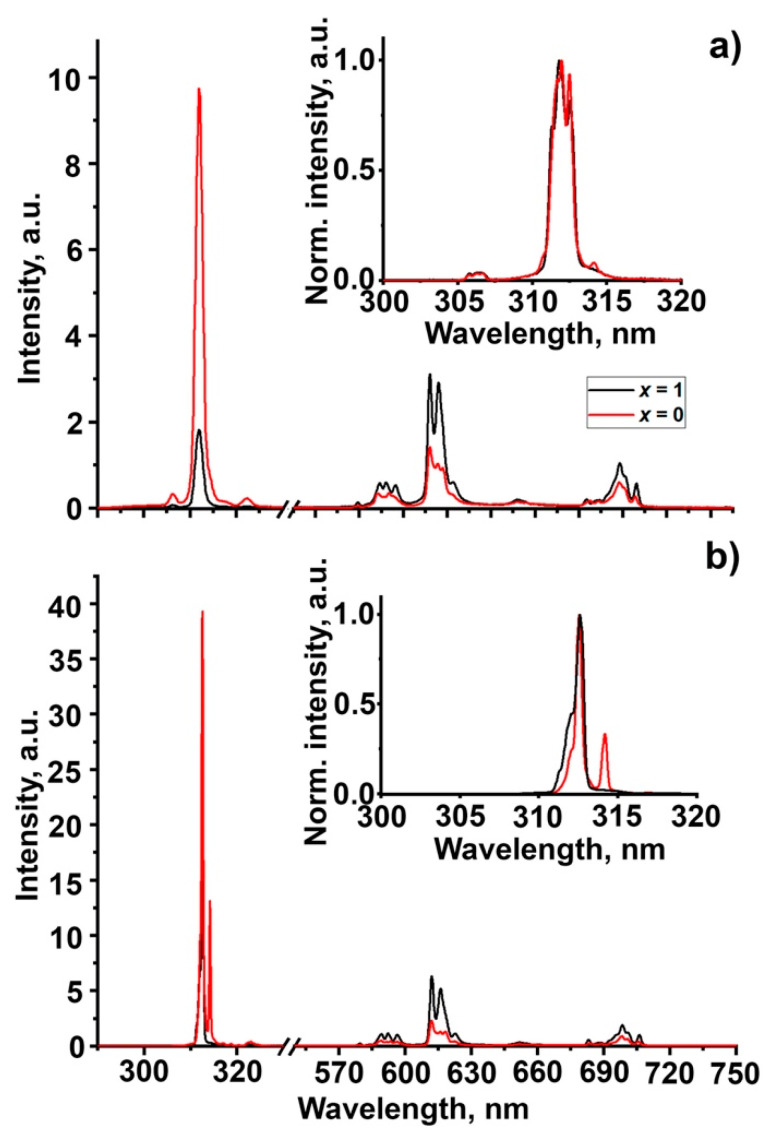
PL spectra of Ca_9-*x*_Zn*_x_*Gd_0.9_Eu_0.1_(PO_4_)_7_ *x* = 0 (red), 1 (black) (*λ*_ex_ = 163 nm) at 300 K (**a**) and 6K (**b**). The inset shows the detailed PL spectra in the 300–320 nm region for both temperatures.

**Figure 6 molecules-28-00352-f006:**
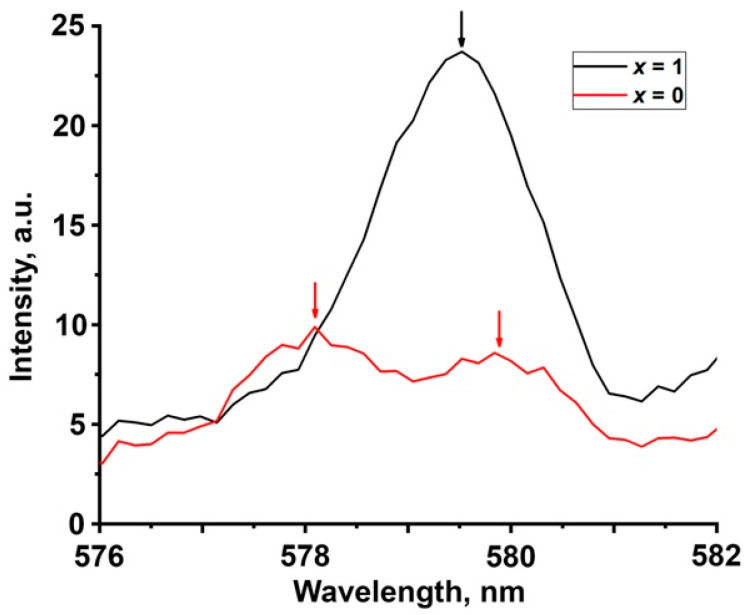
Detailed PL spectra in the 576–582 nm region (^5^D_0–_^7^F_0_ transition in Eu^3+^) measured at 6 K for Ca_9−*x*_Zn*_x_*Gd_0.9_Eu_0.1_(PO_4_)_7_ *x* = 0 (red), 1 (black).

**Figure 7 molecules-28-00352-f007:**
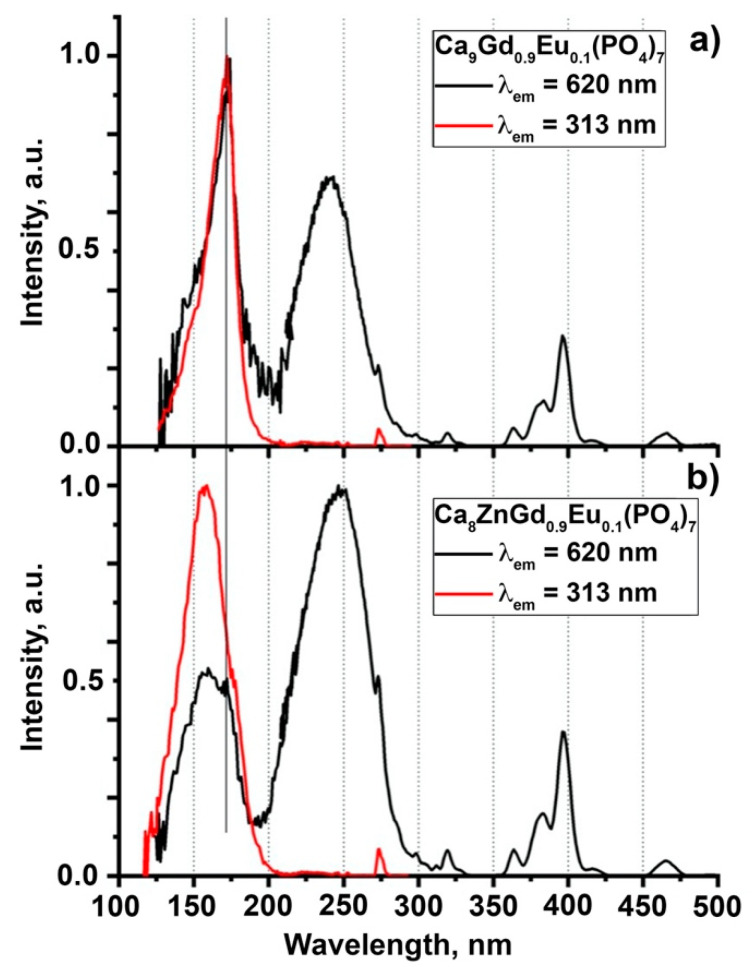
Normalized PLE spectra of Eu^3+^ (*λ*_em_ = 620 nm, black), Gd^3+^ (*λ*_em_ = 313 nm, red) in Ca_9−*x*_Zn*_x_*Gd_0.9_Eu_0.1_(PO_4_)_7_ *x* = 0 (**a**), 1 (**b**).

**Figure 8 molecules-28-00352-f008:**
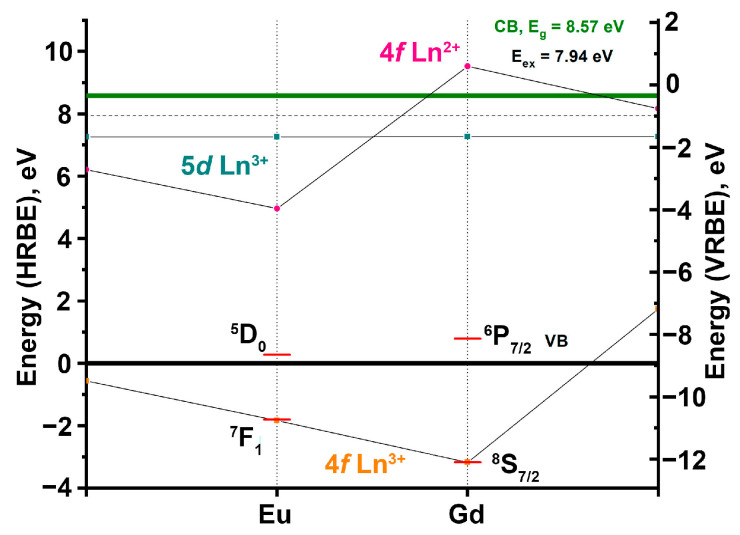
Energy position of the ground and excited levels of 4*f Ln*^3+^ (orange squares), ground levels 5*d Ln*^3+^ (cyan squares), 4*f Ln*^2+^ (pink dots) for Ca_8_Zn*Ln*(PO_4_)_7_ (*Ln*^3+^ = Gd, Eu). Lower black bold line corresponds to the top of the valence band (VB), upper green bold line—bottom of the conduction band (CB). Dashed line near CB—the exciton formation energy (E_ex_).

**Table 1 molecules-28-00352-t001:** EDX analysis results, estimated crystalline size and average full width at half maximum (FWHM) for 0 2 10 (hkl) reflection at 2θ~31° (CuK*_a_*_1_K*_a_*_2_ radiation) of the diffraction peaks for Ca_9−*x*_Zn*_x_*Gd_0.9_Eu_0.1_(PO_4_)_7_.

x	Ca, at.%	Zn, at.%	Gd, at.%	Eu, at.%	Zn:Ca ratio	Crystalline sizes, nm	FWHM, °
0.35	87.29 ± 0.29	3.19 ± 0.16	8.80 ± 0.19	0.72 ± 0.11	0.32 ± 0.03:8.73 ± 0.11	75 ± 10	0.130 ± 0.013
0.50	83.56 ± 0.50	5.74 ± 0.23	9.90 ± 0.18	0.80 ± 0.10	0.57 ± 0.02:8.36 ± 0.05	103 ± 8	0.104 ± 0.006
0.75	81.39 ± 0.46	8.34 ± 0.37	9.44 ± 0.20	0.83 ± 0.13	0.83 ± 0.04:8.14 ± 0.05	108 ± 10	0.101 ± 0.006
1.00	78.25 ± 0.14	11.12 ± 0.22	9.65 ± 0.18	0.98 ± 0.08	1.11 ± 0.02:7.83 ± 0.01	270 ± 20	0.072 ± 0.002

**Table 2 molecules-28-00352-t002:** Crystallographic data for Ca_8_ZnGd(PO_4_)_7_ (SG *R*$\bar{3}$ *c*, Z = 6, T = 293 K).

Sample Composition	Ca_8_ZnGd(PO_4_)_7_
Lattice parameters: *a*, Å	10.3796(6)
*c*, Å	37.1316(3)
Unit cell volume, Å^3^	3460.69(4)
Calculated density, g/cm^3^	3.452(2)
**Data Collection**	
Diffractometer	Empyrean X-ray
Radiation/wavelength (λ, Å)	CuK_α_/1.540593, 1.544427
2θ range (^o^)	10.013–109.983
Step scan (2θ)	0.013
*I* _max_	51281
Number of points	7691
**Refinement**	
Refinement	Rietveld
Background function	Pseudo-Voigt, 16 terms
No. of reflections (all/observed)	488/473
No. of refined parameters/refined atomic parameters	64/54
*R* and *R*_w_ (%) for Bragg reflection (*R*_all_/*R*_obs_)	8.81/9.07, 9.10/9.08
*R*_P_, *R*_wP_, *R*_exp_ (%)	4.73, 6.64, 1.96
Max/min residual density (e) (Å^3^)	1.58/−1.78
n*_f_*_-Ca_*M*1	1.263(8)
n*_f_*_-Ca_*M*3	0.605(5)
n*_f_*_-Zn_*M*5	0.864(8)

## Data Availability

Not applicable.

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
