# Peer review of "New Series of Red-Light Phosphor Ca9−xZnxGd0.9(PO4)7:0.1Eu3+ (x = 0–1)"

_molecules, 2023, doi:10.3390/molecules28010352_

Round 1
Reviewer 1 Report
The article by D.А. Spassky et al is devoted to the design and study of a series of phosphors Ca9-xZnxGd0.9(PO4)7:0.1Eu3+ (x = 0.00 – 1.00) for a red-light phosphor. The paper is comprehensive and the work seems to be well done. It provides a possible roadmap for the development of Eu-based red-light phosphor. Overall, the article is certainly of interest, and may be published after a major revision.
1. Firstly, as the title “New series of red-light phosphor Ca9-xZnxGd0.9(PO4)7:0.1Eu3+ (x = 0 ÷1 )”, what’s the “÷” mean? Is it should be the “-”?
2. The main remarks are related to the characterization of the material. From PXRD pattern, the compound seems to be single-phase, but there is no certainty. What are the refined factors of the series of phosphors Ca9-xZnxGd0.9(PO4)7:0.1Eu3+ (x = 0.00 – 1.00) sample, can you give (such as Rwp(%), Rp(%) and χ2) ?
3. Does the Ca, Zn, Gd, Eu content found by the ICP MS method? Is it agree with the calculated one?
4. Some more references should be added to the list of references to other works on red-light phosphor, for example, DOI: 10.1016/j.jlumin.2018.05.015; 10.1007/s11664-018-6532-y; 10.1007/s10854-019-02283-z; 10.3390/ma15227933.
5. The language should be carefully checked and improved: For example, in conclusion “Obtained in this work solid solutions with the β-Ca3(PO4)2-type structures do not contain impurity phases and all crystallize in R3c or R3c SGs depending on the content of Zn2+ ions.” should be modified to “The as-obtained solid solutions with the β-Ca3(PO4)2-type structures in this work do not contain impurity phases and all crystallize in R3c or R3c SGs depending on the content of Zn2+ ions”.
Author Response
26 December 2022
Dr. Dennis Zhou,
Assistant Editor
Molecules
Dear Dr. Dennis Zhou,
We are grateful to the reviewers for their positive and constructive comments on our manuscript No. molecules-2104332 titled: “New series of red-light phosphor Ca9-xZnxGd0.9(PO4)7:0.1Eu3+ (x = 0 - 1)”. The manuscript has been revised according to the reviewers’ suggestions. The changes, which have been made are listed below along with our response to the reviewers. Revisions to the manuscript marked up using “Track Changes” function of MS Word.
Reviewer 1
Comment 1. Firstly, as the title “New series of red-light phosphor Ca9-xZnxGd0.9(PO4)7:0.1Eu3+ (x = 0 ÷1)”, what’s the “÷” mean? Is it should be the “-”?
Response 1. We changed symbol in the title of the manuscript as suggested.
Comment 2. The main remarks are related to the characterization of the material. From PXRD pattern, the compound seems to be single-phase, but there is no certainty. What are the refined factors of the series of phosphors Ca9-xZnxGd0.9(PO4)7:0.1Eu3+ (x = 0.00 – 1.00) sample, can you give (such as Rwp(%), Rp(%) and χ2)?
Response 2. We did not refine the crystal structure of Ca9-xZnxGd0.9(PO4)7:0.1Eu3+ (x = 0.00 – 1.00). Le-Bail analysis was performed using JANA2006 software package.
Comment 3. Does the Ca, Zn, Gd, Eu content found by the ICP MS method? Is it agree with the calculated one?
Response 3. We performed EDX analysis for five samples (x = 0.35; 0.50; 0.75; 1.00) of the Ca9-xZnxGd0.9(PO4)7:0.1Eu3+ series. The results show insignificant deviation from the theoretical composition. The corresponding data were added to the manuscript (Table 1).
Comment 4. Some more references should be added to the list of references to other works on red-light phosphor, for example, DOI: 10.1016/j.jlumin.2018.05.015; 10.1007/s11664-018-6532-y; 10.1007/s10854-019-02283-z; 10.3390/ma15227933.
Response 4. Thank you for the proposed references. We have carefully read the corresponding articles and introduced two references from the list proposed by reviewer in to the manuscript as refs 32-33. Two other proposed references devoted to “A facile synthesized Eu‐based metal–organic frameworks sensor for highly selective detection of volatile organic compounds” and “A Novel Dual-Emission Fluorescence Probe Based on CDs and Eu3+ Functionalized UiO-66-(COOH)2 Hybrid for Visual Monitoring of Cu2+” do not contain the data which could be related to the topic of this study and therefore we do not include them into the manuscript.
Comment 5. The language should be carefully checked and improved: For example, in conclusion “Obtained in this work solid solutions with the β-Ca3(PO4)2-type structures do not contain impurity phases and all crystallize in R3c or R3c SGs depending on the content of Zn2+ ions.” should be modified to “The as-obtained solid solutions with the β-Ca3(PO4)2-type structures in this work do not contain impurity phases and all crystallize in R3c or R3c SGs depending on the content of Zn2+ ions”.
Response 5. We checked language and made some changes as suggested.
We hope that the revised manuscript will be suitable for publication.
Sincerely yours,
Dr. Dmitry Spassky,
Institute of Physics, University of Tartu, W. Ostwald str. 1, 50411 Tartu, Estonia
E-mail: daspassky@gmail.com

Reviewer 2 Report
In their manuscript the authors investigate the crystallographic and luminescence properties of Zn2+ substituted, Eu3+ doped Ca9Gd(PO4)7. The manuscript is well written but has some inaccuracies. Also, I’m not sure if it fits to the scope of the journal, however, the editor has to decide on this. Language should be checked.
Please find more detailed remarks below. I advise a minor revision.
The role of Zn2+ is not fully explained. Zn2+ is also photoactive. It usually shows absorption and luminescence emission in the near UV. This is actually seen in Figs. 4-6. Zn2+ incorporation notably increases Gd3+ and Eu3+ luminescence, most probably by absorption and subsequent energy transfer to both rare earth ions. This should be added to the manuscript.
p 7, l 217: 575-520 nm should be 575-720 nm
p 7: The reason the transition 5D0-7F0 can indicate the number of crystallographic positions is that this transition is not split by the crystal field. Both levels involved do not split up, so there usually is only one peak in any material. Of course the number of peaks only indicates the minimum of different sites. If the sites are comparably similar still only one peak is observed, since both peaks are superimposed. The sentence in line 221 should be changed accordingly “The number of peaks correlates with the number of cationic positions, however, different crystallographic sites can result in similar peak positions...”
Fig. 6: There is no emission at 308 nm. Please check the number. Emission is at slightly higher wavelengths.
PLE spectra: The broad band between 200 and 300 nm is indeed usually attributed to a charge transfer transition, but from the oxygen ligands to the Eu3+ ion. I don’t think, there is Eu2+ in that composition. If you assume Eu2+, it should be proven. Zn2+ absorption could also be in this region, as explained earlier. Comparison between Figs. 6a and 6b supports this. Investigation of an undoped sample would also be useful here. The band below 200 nm could be due to VB absorption. The position of the VB can be calculated if the crystal structure is known... Probably a literature research could help here. The attribution of these peaks to excitons is unlikely and must be referenced. Also the 4f-5d absorption of Eu3+ could be found in that spectral region below 200 nm. Again, a sample without Eu-doping would help here to clarify the origin of these bands. The whole paragraph must be edited before publication.
p 5: I struggle with the definition of the distortion DI. Different bond lengths are not necessarily an indicator of distorted sites, however I’m no crystallography specialist. Give references if this is a widely used formula.
Author Response
26 December 2022
Dr. Dennis Zhou,
Assistant Editor
Molecules
Dear Dr. Dennis Zhou,
We are grateful to the reviewers for their positive and constructive comments on our manuscript No. molecules-2104332 titled: “New series of red-light phosphor
Ca9-xZnxGd0.9(PO4)7:0.1Eu3+ (x = 0 - 1)”. The manuscript has been revised according to the reviewers’ suggestions. The changes, which have been made are listed below along with our response to the reviewers. Revisions to the manuscript marked up using “Track Changes” function of MS Word.
Reviewer 2
Comment 1. Although the authors mainly focus on structural studies, it would be interesting to find out the results of the synthesis. What were the products of the synthesis? It would also be useful to provide SEM images of the synthesized phases and EDX analysis data for the reader's benefit.
Response 1. The product of the synthesis (besides of main product -
Ca9-xZnxGd0.9(PO4)7:0.1Eu3+) were CO2 and H2O. The reaction is now presented in the “Sample preparation” part. The SEM images were obtained for samples x = 0.00; 0.25; 0.50; 0.75; 1.00 and are presented in Fig. 2 of the manuscript. EDX analysis data are presented in Table 1.
Comment 2. Though there are many studies on Ca3(PO4)2 materials, I strongly recommend providing a crystal structure of the compound, where all cation positions will be labeled. For example, in the introduction section.
Response 2. We have added a structure projection to the introduction section (Fig. 1).
Comment 3. The authors mentioned that the routine XRD experiment is not suitable for separation of centrosymmetric and noncentrosymmetric phases. How did the authors determine that both phases exist in the range of x = 0.35 - 0.75?
Response 3. We determine this according to the results of crystalline size analysis, which were calculated by the Debye-Scherrer equation (Table 1). The broadening of the diffraction peaks and the decrease of crystalline sizes in this region indicate the coexistence of the two phases. In addition, according to the law of phases, there should be a two-phase region between two regions of solid solutions with symmetry R3c and R-3c.
Comment 4. Did the authors estimate the ratio of both phases?
Response 4. We didn’t estimate the ratio of both phases. In routine laboratory PXRD experiment it can’t be possible to determine each of them - Bragg reflections are indistinguishable and merge with each other due to the proximity of the unit cells parameters.
We hope that the revised manuscript will be suitable for publication.
Sincerely yours,
Dr. Dmitry Spassky,
Institute of Physics, University of Tartu, W. Ostwald str. 1, 50411 Tartu, Estonia
E-mail: daspassky@gmail.com

Reviewer 3 Report
Dikhtyar et al. presented results on synthesis and structural characterization of a new series of phosphors Ca9-xZnxGd0.9(PO4)7:0.1Eu3+, consisting of centro- and noncentrosymmetric phases with beta- Ca3(PO4)2-type structure. The subject and object of this study are fully consistent with the aims and scopes of the journal. The research is actual and will make a good contribution to modern materials science. Nevertheless, I have a few comments (see below).
1. Although the authors mainly focus on structural studies, it would be interesting to find out the results of the synthesis. What were the products of the synthesis? It would also be useful to provide SEM images of the synthesized phases and EDX analysis data for the reader's benefit.
2. Though there are many studies on Ca3(PO4)2 materials, I strongly recommend providing a crystal structure of the compound, where all cation positions will be labeled. For example, in the introduction section.
3. The authors mentioned that the routine XRD experiment is not suitable for separation of centrosymmetric and noncentrosymmetric phases. How did the authors determine that both phases exist in the range of x = 0.35 - 0.75?
4. Did the authors estimate the ratio of both phases?
Author Response
26 December 2022
Dr. Dennis Zhou,
Assistant Editor
Molecules
Dear Dr. Dennis Zhou,
We are grateful to the reviewers for their positive and constructive comments on our manuscript No. molecules-2104332 titled: “New series of red-light phosphor
Ca9-xZnxGd0.9(PO4)7:0.1Eu3+ (x = 0 - 1)”. The manuscript has been revised according to the reviewers’ suggestions. The changes, which have been made are listed below along with our response to the reviewers. Revisions to the manuscript marked up using “Track Changes” function of MS Word.
Reviewer 3
Comment 1. The role of Zn2+ is not fully explained. Zn2+ is also photoactive. It usually shows absorption and luminescence emission in the near UV. This is actually seen in Figs. 4-6. Zn2+ incorporation notably increases Gd3+ and Eu3+ luminescence, most probably by absorption and subsequent energy transfer to both rare earth ions. This should be added to the manuscript.
Response 1. Zn2+ plays an essential role in the enhancement of the luminescent properties of whitlokite type phosphates. However, the role of Zn2+ in formation of additional energy transfer channels to Eu3+ and/or Gd3+ is not obvious. It is possible role can be analyzed using the luminescence excitation spectra presented in Fig. 7 (in the new numeration) where excitation spectra of Eu3+ and Gd3+ luminescence are presented for two phosphates – with and without Zn in its composition. One can see that in the region of UV (and, in particular in near UV) the excitation spectra have no noticeable difference for compounds with and without Zn. For both compounds the excitation spectra consists of a set of narrow lines related to 4f-4f transitions in Eu3+ (in the region of 320 – 500 nm) and Gd3+ (at 273 nm) and a broad band peaking at 245 nm, which is relayed to charge transfer 2pO-4fEu transitions. No additional bands, which could allow to make conclusions on the role of Zn2+ in the formation of additional energy transfer channels, has been observed in the excitation spectra of phosphate with Zn. Even in the excitation spectra of Gd3+ where charge transfer band intensity is very low no additional features has been found. To our opinion it indicates that Zn2+ does not form additional energy transfer channels in the studied compounds. The role of Zn2+ has been previously discussed in recent papers [10.1016/j.jallcom.2019.152352 ; 10.1016/j.jallcom.2022.164521]. According to these studies phosphates with Zn2+ demonstrate enhanced luminescent intensity due to the following factors: 1) shifting of the M3 position from the third-order axis; 2) distortion of M3O8 polyhedra (local decrease of the symmetry); 3) general increase in symmetry of the structure (R3c ® Rc).
Comment 2. p 7, l 217: 575-520 nm should be 575-720 nm
Response 2. Thank you for pointing out this inaccuracy. We made changes as suggested.
Comment 3. p 7: The reason the transition 5D0-7F0 can indicate the number of crystallographic positions is that this transition is not split by the crystal field. Both levels involved do not split up, so there usually is only one peak in any material. Of course the number of peaks only indicates the minimum of different sites. If the sites are comparably similar still only one peak is observed, since both peaks are superimposed. The sentence in line 221 should be changed accordingly “The number of peaks correlates with the number of cationic positions, however, different crystallographic sites can result in similar peak positions...”
Response 3. We made changes as suggested. Two peaks attributed to the 5D0-7F0 transition were observed for the sample with x = 0, while only one – for x = 1. The number of peaks correlate with decreasing in number of cationic positions, however, different crystallographic sites with similar crystallographic properties can result in superimposed peak.
Comment 4. Fig. 6: There is no emission at 308 nm. Please check the number. Emission is at slightly higher wavelengths.
Response 4. The excitation spectra for Gd3+ was measured at lem = 313 nm. Thank you for pointing out this inaccuracy. We made changes as suggested (Fig. 7 in the new numeration).
Comment 5a. PLE spectra: The broad band between 200 and 300 nm is indeed usually attributed to a charge transfer transition, but from the oxygen ligands to the Eu3+ ion. I don’t think, there is Eu2+ in that composition. If you assume Eu2+, it should be proven.
Response 5a. The most familiar type of electron or charge transfer (CT) known in luminescence phosphors is that of an electron from the VB to Eu3+. Thus, after the transfer of the electron to Eu3+ it becomes Eu2+. The final state of charge transfer transition is the 4f7 ground state of Eu2+ together with a hole left on the neighboring anion ligands. This process can be described by the equation VB + hn = Eu2+/3+. After excitation the lattice will relax on the ps time scale and upon the back transfer of the electron to the hole on the ligands, sufficient energy is available to leave Eu3+ in an excited state which is then followed by the 5D0;1;2;3 ® 7FJ Eu3+ emissions [http://dx.doi.org/10.1016/j.optmat.2017.03.061].
Comment 5b. Zn2+ absorption could also be in this region, as explained earlier. Comparison between Figs. 6a and 6b supports this.
Response 5b. Here we present the excitation spectra from Fig. 6 (in the initial version of manuscript; Fig. 7 in revised version of manuscript) in a different way for better visibility (see below). The excitation spectra of Eu3+ luminescence (left panel) and Gd3+ luminescence (right panel) of the compounds with and without Zn2+ are similar in UV spectral region. It indicates that role of possible existing energy transfer channels connected with Zn2+ is negligible for UV spectral region. Excitation spectra demonstrate differences only in the VUV region, which will be discussed below.
Figure. Normalized PLE spectra of Eu3+ (lem = 620 nm, black, left panel), Gd3+ (lem = 313 nm, red, right panel) in Ca9-xZnxGd0.9Eu0.1(PO4)7 x = 0 (lower row) and 1 (upper row).
Comment 5c. Investigation of an undoped sample would also be useful here. The band below 200 nm could be due to VB absorption. The position of the VB can be calculated if the crystal structure is known...
Response 5c. The study of the “undoped” sample has been performed by us previously in [10.1016/j.jallcom.2022.164521]. In this paper we carefully checked the excitation spectra of similar compounds Ca8ZnLn(PO4)7 with Ln3+ = La, Ce, Pr, Nd, Tb, Gd, Yb, Dy, Tm, Sm, Eu, Lu (see figure below). For the samples with non-luminescent Ln3+ = La and Lu ions the intrinsic emission has been detected and the excitation spectra of this emission were used for the determination of the energy of the direct creation of excitons (Eex =7.94 eV) as well as bandgap energy (Eg =8.57 eV). The previously obtained Eex value was also used in the analysis of the results presented in the manuscript. It is shown that the Eex value obtained previously corresponds to that obtained in the present study (Eex =7.89 eV). Also, calculation of the VB top position has been performed in [10.1016/j.jallcom.2022.164521] and the scheme with the vacuum referred binding energy (VRBE) of VB top was presented in reference and this data used for the scheme plotted in current manuscript (Fig. 8 in revised version of manuscript).
Comment 5d. Also the 4f-5d absorption of Eu3+ could be found in that spectral region below 200 nm. Again, a sample without Eu-doping would help here to clarify the origin of these bands. The whole paragraph must be edited before publication.
Response 5d. We agree that 4f-5d transitions in Eu3+ may be observed in side bandgap compounds as a set of sharp peaks in VUV spectral region [see, e.g. 10.1103/PhysRevB.65.045113]. The intense sharp excitation peak below the fundamental absorption range is indeed observed with maximum at ~172 nm in Ca9Gd0.9Eu0.1(PO4)7 and it could be related to such transition. However, this sharp peak is intensive in the excitation spectra for both – Gd3+ and Eu3+ luminescence. If this peak is really connected with 4f-5d transitions in Eu3+ then it indicates efficient energy transfer from the Eu3+ to Gd3+. However, according to the analysis of the excitation spectra in UV spectral region the energy transfer is inefficient (CTB is barely observed in the excitation spectra of Gd3+ emission). Therefore, the assumption that the sharp peak at 172 nm is associated with 4f-5d transitions is unlikely to be correct. We further speculate on the origin of this peak as an exciton bounded near the Gd3+ ions. Concluding, we have revised the whole paragraph taking into account the remarks of reviewer and in accordance with the present responses.
Comment 6. p 5: I struggle with the definition of the distortion DI. Different bond lengths are not necessarily an indicator of distorted sites, however I’m no crystallography specialist. Give references if this is a widely used formula.
Response 6. We made a new reference as suggested (ref. 51).
We hope that the revised manuscript will be suitable for publication.
Sincerely yours,
Dr. Dmitry Spassky,
Institute of Physics, University of Tartu, W. Ostwald str. 1, 50411 Tartu, Estonia
E-mail: daspassky@gmail.com

Round 2
Reviewer 1 Report
Based on the revision, the revised manuscript now can be published in Molecules.
Reviewer 3 Report
The reviewers' responses seem to be mixed up as I see comments on other questions. However, judging by the corrected version of the article, the authors have made corrections to my comments as well. I think the paper can be published in the journal.